# Progressive Thinning of Retinal Nerve Fiber Layer/Ganglion Cell Layer (RNFL/GCL) as Biomarker and Pharmacological Target of Diabetic Retinopathy

**DOI:** 10.3390/ijms241612672

**Published:** 2023-08-11

**Authors:** Gianpaolo Zerbini, Silvia Maestroni, Ilaria Viganò, Andrea Mosca, Renata Paleari, Daniela Gabellini, Silvia Galbiati, Paolo Rama

**Affiliations:** 1Complications of Diabetes Unit, Diabetes Research Institute, IRCCS Ospedale San Raffaele, 20132 Milan, Italy; silvia.maestroni@gmail.com (S.M.); vigano.ilaria@hsr.it (I.V.); gabellini.daniela@hsr.it (D.G.); galbiati.silvia@hsr.it (S.G.); 2Centro per la Riferibilità Metrologica in Medicina di Laboratorio (CIRME), Dipartimento di Fisiopatologia Medico-Chirurgica e dei Trapianti, Università degli Studi di Milano, 20133 Milan, Italy; andrea.mosca@unimi.it (A.M.); renata.paleari@unimi.it (R.P.); 3Istituto di Tecnologie Biomediche, Consiglio Nazionale delle Ricerche (ITB-CNR), 20054 Milan, Italy; 4Cornea and Ocular Surface Unit, IRCCS Ospedale San Raffaele, Vita-Salute San Raffaele University, 20132 Milan, Italy; p.rama@smatteo.pv.it

**Keywords:** diabetic retinopathy, prevention, biomarker, optical coherence tomography

## Abstract

Diabetes-driven retinal neurodegeneration has recently been shown to be involved in the initial phases of diabetic retinopathy, raising the possibility of setting up a preventive strategy based on early retinal neuroprotection. To make this possible, it is crucial to identify a biomarker for early retinal neurodegeneration. To this end, in this study, we verified and confirmed that, in the Akita mouse model of diabetes, the thinning of the retinal nerve fiber layer/ganglion cell layer (the RNFL/GCL—the layer that contains the retinal ganglion cells) precedes the death of these same cells, suggesting that this dysfunction is a possible biomarker of retinal neurodegeneration. We then confirmed the validity of this assumption by starting a neuroprotective treatment (based on nerve growth factor eye drops) in concert with the first demonstration of RNFL/GCL thinning. In this way, it was possible not only to avoid the loss of retinal ganglion cells but also to prevent the subsequent development of the microvascular stage of diabetic retinopathy. In conclusion, in the case of diabetes, the thinning of the RNFL/GCL appears to be both a valid biomarker and a pharmacological target of diabetic retinopathy; it precedes the development of vascular dysfunctions and represents the ideal starting point for prevention.

## 1. Introduction

Diabetic retinopathy (DR) is presently one of the major worldwide causes of loss of sight and despite huge research efforts, there is still no effective preventive or medical approach for this complication of diabetes [1]. Treatment, which substantially consists of laser photocoagulation or intravitreal injections of anti-VEGF molecules, is restricted to the final proliferative stage of DR [2,3]. This is mostly due to the fact that, until very recently, the pathophysiology of the complication and, in particular, of its early stages, has been poorly understood.

The recent demonstration that the microvascular stages of DR and, in particular, microaneurysms (commonly considered the first signs of DR) are often preceded by the degeneration of the neuroretina and, in particular, of the retinal ganglion cells (RGCs) [4,5,6,7] has, at the same time, allowed us to clarify the early pathophysiology of the complication and suggest neuroprotection as a reasonable approach for preventing the development of DR [8,9].

Among the established neuroprotective agents, nerve growth factor (NGF), an endogenous neuropeptide that is part of the neurotrophin family [10], appears to be of particular interest. NGF, in fact, plays a key trophic and differentiative role in the neurons of the peripheral and central nervous systems [11]. The intra-brain administration of NGF was shown to have beneficial effects in patients affected by Parkinson’s and Alzheimer’s diseases (A), while the intraocular injection of NGF in an animal model of the optic nerve section was shown to prevent retinal ganglion cell degeneration [12]. The only common side effect of NGF treatment observed during clinical trials is pain at the injection site [13].

In this regard, we have recently demonstrated that early treatment (starting from the onset of diabetes at 3 weeks of age) with topical nerve growth factor (NGF) in the Ins2akita (Akita) mouse model of diabetes [14] allows us to prevent retinal neurodegeneration and, in the long run, the development of the microvascular stage of DR, suggesting that early retinal neurodegeneration is directly involved in the pathogenesis of DR and that neuroprotection can prevent the development of DR [15].

A similar preventive strategy that treats all patients with NGF eye drops for their entire life, starting from the onset of diabetes, even if successful in preventing DR, would not be feasible in a clinical setting for a number of reasons: (1) only a subset of diabetic patients develops DR and, in particular, its proliferative stage [16]; (2) the long-term side effects of NGF treatment are still unknown, so only patients at risk of DR should be treated; and (3) the cost of treating all diabetic patients with topical NGF would be extremely high. The problem could be solved through the identification of a biomarker for use in the early identification of patients at risk of developing diabetes-driven retinal neurodegeneration and, subsequently, DR [17,18].

In our previous study, we noticed that in placebo-treated Akita mice, the first sign of neurodegeneration was represented by a significant thinning of the retinal nerve fiber layer/ganglion cell layer (RNFL/GCL) [15]. At this time point, the number of RGCs was still normal in these animals (significant RGC death would become detectable only later on), suggesting that the thinning of the RNFLGCL may precede RGC death and, for this reason, could represent a valid biomarker as it reasonably indicates not only that the RGCs are suffering, but also that they are still alive. The aim of the present study was, therefore, to verify whether the secondary prevention of DR could still be possible by starting NGF topical treatment after the detection of a significant thinning of the RNFL/GCL.

## 2. Results

To verify whether the early development of an OCT-detectable thinning of the RNFL/GCL could be implemented at the same time as a biomarker of diabetic retinopathy and as an effective pharmacological target, we tested the possibility of preventing (secondary prevention) the development of DR by starting neuroprotective treatment (topical NGF) as soon as the thinning of the RNFL/GCL becomes statistically significant.

In the rat retina, it has been found that streptozotocin-induced diabetes causes programmed RGC death via the upregulation of NGF receptor p75 and that NGF prevents death by normalizing the dysfunction; this finding has been already published in the seminal paper of Hammes et al. [19]. In this paper, we confirm (Figure 1) that the early (6 weeks of age) diabetes-driven upregulation of NGF receptor p75 also characterizes the retinas of Akita mice (ratio p75/tubulin in Akita mice: 1.9 ± 0.3 mean ± SE vs. wild-type mice: 1.0 ± 0.2, *p* = 0.03), thus justifying the subsequent use of NGF (topical treatment in our case) for validating the thinning of the RNFL/GCL as a biomarker and pharmacological target of diabetic retinopathy.

The concentration of glycated hemoglobin was significantly increased in Akita mice and was independent of NGF treatment as shown in Figure 2.

In particular, glycated hemoglobin levels were significantly higher in placebo-treated Akita mice (87.0 ± 4.2 mmol/mol, mean ± SE) compared to placebo-treated wild-type mice (26.0 ± 4.1, *p* < 0.0001) and in NGF-treated Akita mice (88.4 ± 4.2) compared to NGF-treated wild-type mice (29.0 ± 4.1, *p* < 0.0001), but glycated hemoglobin levels did not differ between the two groups of wild-type animals and between the two groups of Akita mice.

As shown in Figure 3A, at eight weeks of age, the RNFL/GCL retinal layer was already significantly thinner in the subgroup of Akita mice that were, since then, treated with placebo (20.1 ± 0.4 µm, mean ± SE) compared to the subgroup of wild-type mice that were, since then, treated with placebo (22.7 ± 0.4, *p* = 0.0004).

Over time, RNFL/GCL remained constantly thinner in placebo-treated Akita mice compared to placebo-treated wild types (18.4 ± 0.4 vs. 22.4 ± 0.4, *p* < 0.0001 at 16 weeks; 19.4 ± 0.3 vs. 22.6 ± 0.3, *p* < 0.0001 at 24 weeks; 18.2 ± 0.3 vs. 23.2 ± 0.3, *p* < 0.0001 at 36 weeks) (Figure 3A).

As seen in Figure 3B, at eight weeks of age (just before starting the treatments), the RNFL/GCL thickness of Akita mice that were, since then, treated with NGF (20.9 ± 0.4 µm) was similar to the RNFL/GCL thickness of Akita mice that were, since then, treated with placebo (*p* = NS) and was significantly thinner compared to RNFL/GCL measured in wild-type mice that were, since then, treated with placebo (*p* < 0.02). Of interest, at 16 weeks of age RN, the FL/GCL thickness of NGF-treated Akita mice (20.8 ± 0.4 µm) significantly increased compared to placebo-treated Akita mice (*p* = 0.002) and was not more different from placebo-treated wild-type mice (*p* = NS). At 24 weeks of age, the RNFL/GCL thickness of NGF-treated Akita mice (22.1 ± 0.3 µm) significantly increased compared to placebo-treated Akita mice (*p* < 0.0001) and was similar to placebo-treated wild-type mice (*p* = NS). Finally, at 36 weeks of age, the RNFL/GCL thickness of NGF-treated Akita mice (21.3 ± 0.3 µm) significantly increased compared to placebo-treated Akita mice (*p* < 0.0001) and became thinner than the placebo-treated wild-type mice (*p* = 0.009). The NGF treatment of wild-type animals did not exert any specific effect at any time point compared to placebo-treated wild-type mice (Figure 3B). The representative pictures of OCTs performed at 36 weeks of age in the four groups of animals considered in this study are shown in Figure 4.

The final confirmation that NGF may prevent RGC loss in Akita mice, even when administered after the thinning of RNFL/GCL, was obtained via direct RGC count in the retinas of sacrificed animals. The RNFL/GCL thickness (Figure 3A) at eight weeks of age was already significantly decreased in the Akita mice that were, since then, treated with placebo; the RGC number at this age was similar between these mice (20.1 ± 0.7, mean ± SE) and wild-type mice that were, since then, treated with placebo (22.1 ± 0.9, *p* = NS), as shown in Figure 5A.

At 24 weeks of age, the RGC number significantly reduced in placebo-treated Akita mice (13.1 ± 1.3) compared to placebo-treated wild-type mice (21.0 ± 0.7, *p* < 0.0001). A similar finding was detected at 36 weeks of age when the RGC number of placebo-treated Akita mice (11.6 ± 1.7) reduced compared to placebo-treated wild-type mice (21.8 ± 1.6, *p* = 0.001).

As seen in Figure 5B, at eight weeks of age (just before starting the treatment with NGF), the RGC number of Akita mice that were, since then, treated with NGF (20.7 ± 0.6 µm) was similar to the RGC number of Akita mice that were, since then, treated with placebo (*p* = NS) and was also similar to the RGC number of wild-type mice that were, since then, treated with placebo (*p* = NS). Of interest, at 24 weeks of age, the RGC number of NGF-treated Akita mice (21.3 ± 1.2 µm) significantly increased compared to placebo-treated Akita mice (*p* < 0.0001) and did not differ from placebo-treated wild-type mice (*p* = NS). Finally, at 36 weeks of age, the RGC number of NGF-treated Akita mice (21.1 ± 2.1 µm) significantly increased compared to placebo-treated Akita mice (*p* = 0.004) and was similar to placebo-treated wild-type mice (*p* = NS). As before, topical NGF treatment of wild-type mice did not exert any specific effect (Figure 5B). Representative pictures of immunofluorescence for the RGC nuclear antigen Brn3a performed at 36 weeks of age in the retinal sections of the four groups of animals considered for the study are shown in Figure 6.

Our findings suggest that NGF prevents diabetes-driven retinal neurodegeneration, even when administered after a significant RNFL/GCL thickness reduction. To determine whether the NGF treatment that started after the thinning of RNFL/GCL (secondary prevention) is effective in the vascular stage of DR, we proceeded with trypsin digestion. As shown in Figure 7A, the number of acellular capillaries was significantly higher in placebo-treated Akita mice (78.0 ± 6.5 number/mm^2^, mean ± SE) compared to placebo-treated wild-type mice (29.4 ± 4.4, *p* < 0.0001), NGF-treated wild-type mice (35.5 ± 4.0, *p* < 0.0001), and NGF-treated Akita mice (45.1 ± 6.5, *p* = 0.0002), which was not different from (*p* = NS) the placebo-treated wild-type mice and NGF-treated wild-type mice.

In parallel, as shown in Figure 7B, the number of retinal pericytes significantly reduced in placebo-treated Akita mice (976.9 ± 28.1 number/mm^2^, mean ± SE) compared to placebo-treated wild-type mice (1147.9 ± 35.4, *p* = 0.001), NGF-treated wild-type mice (1096.3.0 ± 34.2, *p* = 0.03), and to NGF-treated Akita mice (1168.2 ± 40.0, *p* = 0.0004), which was not different from (*p* = NS) placebo-treated wild-type mice and NGF-treated wild-type mice. Representative pictures of trypsin digests performed at 36 weeks of age in the retinal sections of the four groups of animals considered for the study are shown in Figure 8.

## 3. Discussion

The results of this study show that in the Akita mouse, an established mouse model of diabetes and diabetic retinopathy [14], the thinning of RNFL/GCL (as measured in vivo via OCT) precedes the loss of RGCs. The other major achievement of this study is the demonstration that, by starting neuroprotective treatment (NGF eye drops) immediately after the detection of RNFL/GCL thinning (when RGCs are still alive), it is still possible to avoid RGC loss, normalize RNFL/GCL thickness, and prevent the development of the microvascular stage of DR. The above-described findings demonstrate, for the first time, the feasibility of a secondary prevention approach for DR.

The finding that, in the case of diabetes, the thinning of RNFL/GCL, as measured via OCT, precedes the loss of RGCs is particularly intriguing, as it suggests that the diabetes-driven process ending with the death of RGCs is not an acute phenomenon, but it is rather protracted over time. Of interest, the loss of RGCs is a dysfunction shared by diabetes and glaucoma [20]. In the case of glaucoma, axonal degeneration was shown to precede RGC body death [21], a phenomenon that, if confirmed in diabetic patients with DR, could easily explain why the in vivo measurement of RNFL/GCL thickness allows the prediction of RGC loss.

Translation of the secondary prevention approach to diabetic patients should be possible, at least from a theoretical point of view. The thinning of RNFL has been in fact demonstrated in a subgroup of diabetic patients and is associated with an increased risk of developing DR [4,6]. Conversely, patients without DR, even after a long duration of type 1 diabetes, are characterized by normal RNFL thickness [22]. To clarify this issue, a clinical trial could be easily set up; neuroprotective treatment could be started immediately after the demonstration of a significant and persistent thinning of RNFL, and the effects of the treatment could be monitored sequentially via OCT and OCTA in a non-invasive way. Of particular interest, retinal neuroprotection has been already demonstrated in humans after topical administration of NGF [23].

During the last few years, a number of pharmacological treatments, besides NGF, have been shown to be protective against the development of DR [24,25,26]. Whatever the case and whatever the drug to be tested is, the important point from this study is that we now have an available biomarker that can be used to define precisely and in a standardized way the best possible moment when to start treatment and how to monitor the efficacy of the treatment itself over time.

In conclusion, the identification of diabetes-driven thinning of RNFL/GCL as a biomarker and pharmacological target of DR could represent a turning point in the prevention of DR. Clinical studies in humans are now mandatory to confirm and expand the results obtained from animal models of diabetes.

## 4. Materials and Methods

**Western blot.** After extraction, the left retinas of untreated, six weeks of age, wild-type (n = 5) and Akita mice (n = 4) were homogenized with a glass potter apparatus in 200 µL of 0.32 M ice-cold sucrose buffer containing HEPES (1.0 mM), MgCl_2_ (1.0 mM), NaHCO_3_ (1.0 mM), NaF (10 mM), 25X protease inhibitors (Complete; Roche Diagnostics, Basel, Switzerland), and 10X phosphatase inhibitors (Sigma-Aldrich, St. Louis, MO, USA). Finally, 2 µL of 10% Triton was added to the homogenized sample. Automated Western immunoblotting was performed using JESS Simple Western™ (ProteinSimple^®^, Bio-Techne, Minneapolis, MN, USA), a fully automated capillary-based system. The samples were diluted to total protein concentrations ranging from 10.0 to 3.0 ng/µL, and 3 µL was added to each well. NGFRp75 mouse monoclonal antibody (Santa Cruz, sc-271708) was used at a 1:10 dilution, and beta-tubulin rabbit polyclonal antibody (Proteintech 10094-1-AP) at a 1:300 dilution. Ten µL of the secondary Ab (anti-mouse and anti-rabbit HRP) was used, according to the reagent’s instructions (anti-mouse and anti-rabbit detection module chemiluminescence; ProteinSimple^®^, Bio-Techne, Minneapolis, MN, USA). The plate and capillary cartridge used were included in the 12–230 kDa separation module (ProteinSimple^®^, Bio-Techne). The results obtained were analyzed using Compass for Simple Western software (version 6.2.0).

**Design of the study**. Four groups of seven animals (males) were studied for 28 weeks (between 8 and 36 weeks of age). We only took into consideration male animals, as Ins2akita females have significantly lower blood glucose levels [27]. The animals were randomly assigned to the four different groups that were topically treated with eye drops (two drops per eye per day starting at eight weeks of age) containing placebo or NGF (180 mg/mL) as follows: (1) wild-type (C57BL6J) mice, placebo-treated, n = 7; (2) Akita mice, placebo-treated, n = 7; (3) wild-type mice, NGF-treated, n = 7; (4) Akita mice, NGF-treated, n = 7. Eye drops of recombinant human NGF (rhNGF, simply called NGF) were provided by Dompé S.p.A. L’Aquila, Italy.

The animals in the different groups were evaluated using optical coherence tomography (OCT) at 8 weeks of age (baseline, just before starting the pharmacological treatment) and then at 16, 24, and 36 weeks of age. The number of RGCs was counted in the left eye using Brn3a staining after sacrificing the animals at the end of the study (36 weeks). RGCs were counted also at 8 and 24 weeks of age. To this end, fifty-six more animals per group were included in the study, divided into four groups, treated with eye drops as described above, and sacrificed at eight weeks of age (28 animals) and at twenty-four weeks of age (28 animals). The number of retinal acellular capillaries and pericytes was determined in the right eye by performing trypsin digestion at 36 weeks of age.

**Optical coherence tomography (OCT)**. The retinas of the animals included in the study were studied by means of the retinal imaging microscope Micron IV together with Image-Guided 830 nm OCT (Phoenix Research Laboratories, Pleasanton, CA, USA), as previously described [15]. Briefly, the animals were anesthetized with an intraperitoneal injection of 80 mg/kg ketamine and 10 mg/kg xylazine (Sigma-Aldrich, Munich, Germany). Mydriasis was induced by eye-drop tropicamide 0.5% (Visumidriatic, Tibilux Pharma, Italy). Hydroxyethyl cellulose (Gel 4000 2%; Bruschettini, Italy) was used to keep the cornea moist. OCT bidimensional scans (B-scan) were obtained by performing a 550 μm diameter circular scan around the head of the optic nerve. Both eyes were studied, and the results were averaged. Insight software (Version 1.0.7, Phoenix Research Laboratories) was used to obtain retinal layer segmentation and quantification.

**Count of retinal ganglion cells (RGCs)**. The number of RGCs was evaluated after Brn3a staining using a goat polyclonal anti-Brn3a (Santa Cruz, Santa Cruz, CA), as previously described [15,28].

**Trypsin Digestion**. Trypsin digestion was performed as previously described [15], following the original methodology set up by Dietrich and Hammes [19,29]. The slides, stained with hematoxylin and eosin, were scanned using Aperio^®^ ePathology digital scanner, and images were analyzed using ImageScope™ software Version 12.1 (both from Leica Biosystems, Nussloch, Germany). Acellular capillaries and pericytes were counted in ten independent fields randomly chosen for each retina and finally corrected for the density of capillaries (number of acellular capillaries/pericytes per mm^2^ of capillary area).

**Glycated Hemoglobin**. Glycated hemoglobin (GHb) was quantified as previously described [15], using an automated HPLC analyzer, based on boronate affinity chromatography (Premier Hb9210, Trinity Biotech, Menarini, Firenze, IT). The analyzer was calibrated and used according to the instructions of the manufacturer. The blood samples were studied and analyzed as haemolysates using specific racks. Two control materials with low and high HbA1c concentrations, specifically supplied by the manufacturer, were used as internal quality controls for each analytical run.

**Statistics**. Data are shown as the arithmetic means ± standard error. Comparisons between groups were performed using ANOVA, and multiple comparisons were performed using the Tukey–Kramer test (JMP software Version 7.0 for the Apple Macintosh; SAS Institute, Cary, NC, USA). The null hypothesis was rejected at the 5% level (two-tailed). Changes over time in blood glucose, body weight, RNFL, and number of RGCs were evaluated using ANOVA for repeated measures.

## Figures and Tables

**Figure 1 ijms-24-12672-f001:**
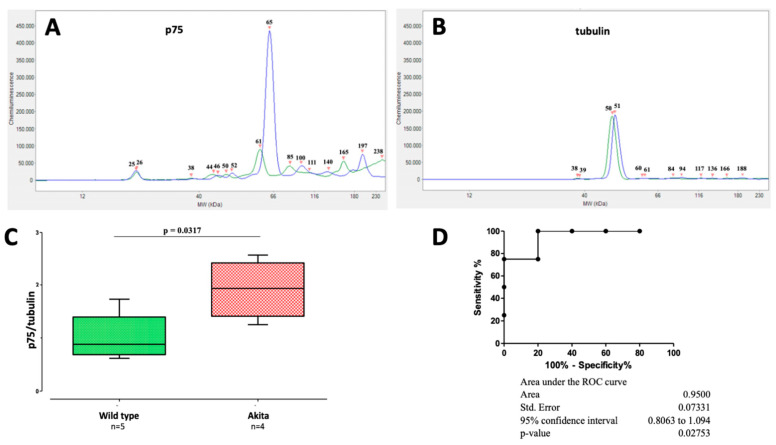
**P75 protein expression in the retinas of Akita and wild-type mice**. An example of a capillary-based assay graph visualization of p75 (**A**) and tubulin (**B**) proteins in the retina of an Akita (blue) and a wild-type mouse (green). Molecular weights of the peaks (MW-kDa) were calculated automatically by comparing them to the reference ladder (12–230-kDa). Images from the high dynamic range 4.0 were used for the analysis. (**C**) Box plots comparing the p75/tubulin ratio levels in wild-type (n = 5) and Akita mice (n = 4), *p* = 0.03. (**D**) ROC curve analysis of the p75/tubulin ratio levels in the retinas of wild-type (n = 5) and Akita mice (n = 4).

**Figure 2 ijms-24-12672-f002:**
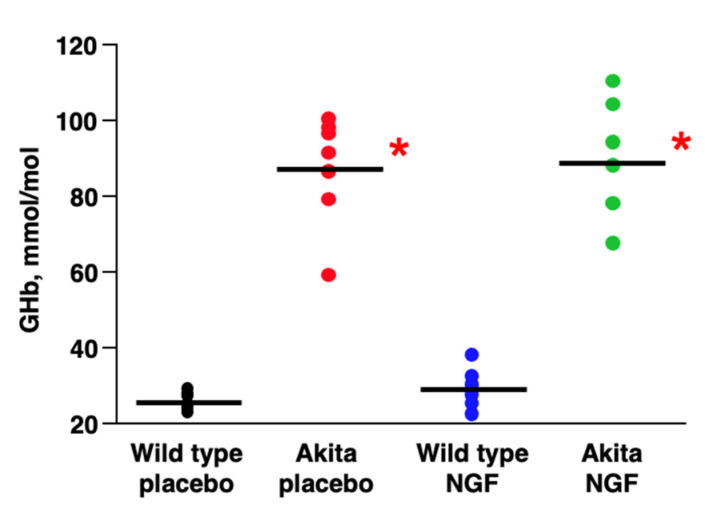
**Glycemic control**. Glycated hemoglobin (GHb) levels at the end of the study (36 weeks) were significantly increased in Akita mice compared to control animals, * *p* < 0.0001. Glycemic control was independent of the treatment.

**Figure 3 ijms-24-12672-f003:**
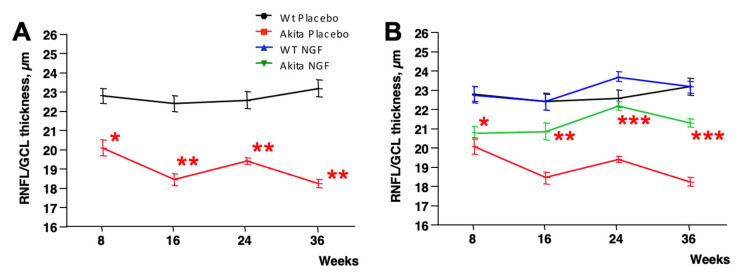
**Time course of the RNFL/GCL layer measured in vivo by OCT**. (**A**) Evolution over time (8–36 weeks) of RNFL/GCL thickness in placebo-treated Akita mice (red line) compared to placebo-treated wild-type mice (black line). Placebo-treated Akita mice showed a significant reduction in thickness (* *p* = 0.004; ** *p* < 0.0001) at 8, 16, 24, and 36 weeks of age. (**B**) Just before starting the topical treatment (eight weeks of age), RNFL/GCL thickness was similar in placebo-treated Akita mice that were, since then, treated with placebo (red line) or with NGF (green line), and in both cases, RNFL/GCL thickness was significantly reduced compared to wild-type mice that were, since then, treated with placebo (black line) (* *p* = 0.02). RNFL/GCL thickness was significantly increased (** *p* = 0.002; *** *p* < 0.0001) at 16, 24, and 36 weeks of age in NGF-treated Akita mice (green line) compared to placebo-treated Akita mice (red line). NGF treatment in wild-type mice (blue line) did not result in any specific effects compared to placebo-treated wild-type mice.

**Figure 4 ijms-24-12672-f004:**
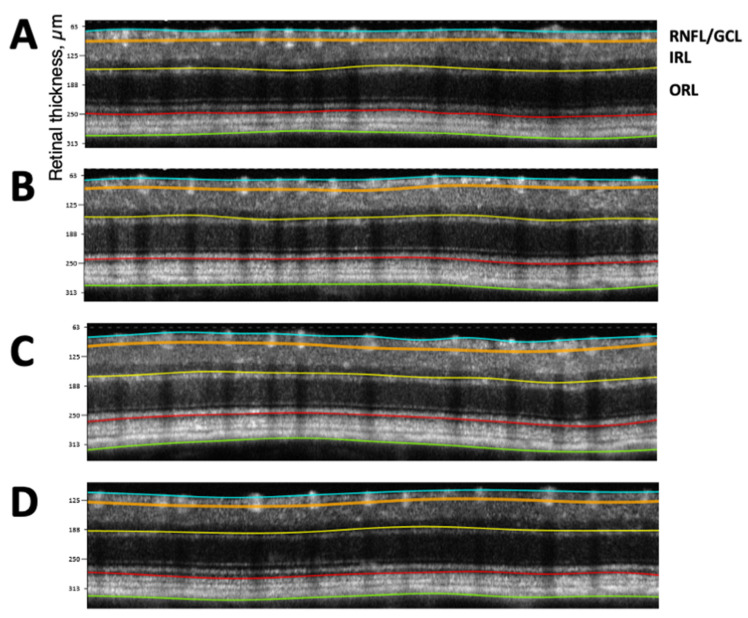
**Representative pictures of optical coherence tomography (OCT)**. Representative pictures of OCTs performed at 36 weeks of age (**A**). Placebo-treated wild-type mice (**B**). Placebo-treated Akita mice (**C**). NGF-treated wild-type mice and (**D**). NGF-treated Akita mice.

**Figure 5 ijms-24-12672-f005:**
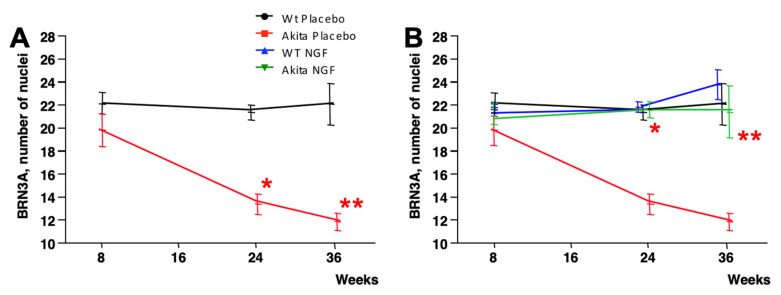
**Time course of the RGC number counted ex vivo after staining for Brn3a**. (**A**) Evolution over time (8–36 weeks) of RGC number in placebo-treated Akita mice (red line) compared to placebo-treated wild-type mice (black line). The number of RGCs was similar between the two groups of animals at 8 weeks of age. Over time, the number of RGCs started to decrease in placebo-treated Akita mice and, at 24 and 36 weeks of age, was significantly reduced compared to placebo-treated wild-type mice (* *p* < 0.0001, ** *p* = 0.001). (**B**) The number of RGCs in NGF-treated Akita mice (green line) remained substantially stable over time and, as a consequence, at both 24 and 36 weeks of age, it significantly increased compared to placebo-treated Akita mice (red line) (* *p* < 0.0001; ** *p* = 0.004). NGF treatment in wild-type mice (blue line) did not exert any specific effect.

**Figure 6 ijms-24-12672-f006:**
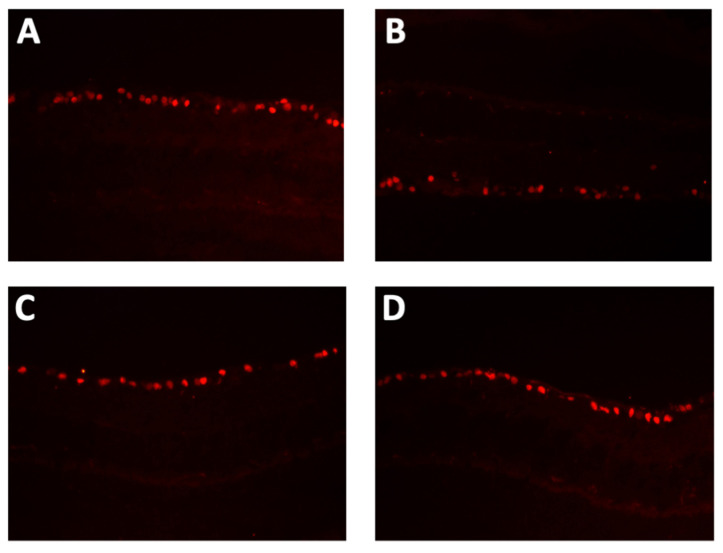
**Representative Brn3a immunostaining pictures**. Representative pictures of immunofluorescence for the RGC nuclear antigen Brn3a performed at 36 weeks of age in the retinal sections of (**A**) placebo-treated wild-type mice, (**B**) placebo-treated Akita mice, (**C**) NGF-treated wild-type mice, and (**D**) NGF-treated Akita mice.

**Figure 7 ijms-24-12672-f007:**
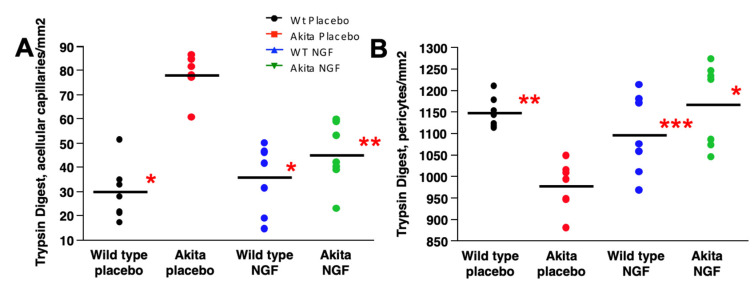
**Count of acellular capillaries and pericytes in the retina**. (**A**). The number of retinal acellular capillaries at the end of the study (36 weeks) was significantly higher (* *p* < 0.0001; ** *p* = 0.0002) in the placebo-treated Akita mice (red dots) compared to placebo-treated wild-type mice (black dots), NGF-treated wild-type mice (blue dots), and NGF-treated Akita mice (green dots). The last three groups were similar to each other (*p* = NS), confirming that NGF treatment normalized the number of retinal acellular capillaries in Akita mice. (**B**). The number of retinal pericytes at the end of the study (24 weeks) was significantly lower (* *p* = 0.0004; ** *p* = 0.001; *** *p* = 0.03) in the placebo-treated Akita mice (red dots) compared to placebo-treated wild-type mice (black dots), NGF-treated wild-type mice (blue dots), and NGF-treated Akita mice (green dots). The last three groups were similar to each other (*p* = NS), confirming that NGF treatment normalized the number of retinal pericytes in Akita mice.

**Figure 8 ijms-24-12672-f008:**
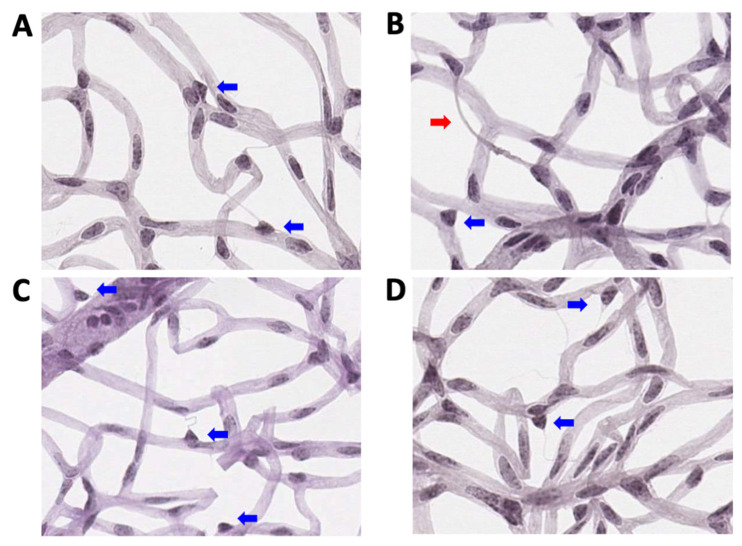
**Representative trypsin-digested pictures**. Representative pictures of trypsin digestion performed at 36 weeks of age in the retinal sections of (**A**) placebo-treated wild-type mice, (**B**) placebo-treated Akita mice, (**C**) NGF-treated wild-type mice, and (**D**) NGF-treated Akita mice. Red arrow indicates an acellular capillary and blue arrows indicate the “triangular” nuclei of pericytes localized outside the capillary wall.

## Data Availability

The raw data of this article will be made available by the authors.

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
