# Peer review of "Progressive Thinning of Retinal Nerve Fiber Layer/Ganglion Cell Layer (RNFL/GCL) as Biomarker and Pharmacological Target of Diabetic Retinopathy"

_ijms, 2023, doi:10.3390/ijms241612672_

Round 1

Reviewer 1 Report

Very important, nice work.

Intruduction: Introduce NGF: biochemistry, effect, side effect of treatment

Discussion L236-42: What might explain that axonal damage precedes ganglion cell death? What human studies support the results found in animal studies: RNFL loss precedes reduction in GCL thickness?

Author Response

Reviewer 1 - Point by point response

  1. Intruduction: Introduce NGF: biochemistry, effect, side effect of treatment

A new paragraph (page 2, lines 44-51) and 4 more references (10-13) have been added to the manuscript to introduce NGF

  1. Discussion L236-42: What might explain that axonal damage precedes ganglion cell death? What human studies support the results found in animal studies: RNFL loss precedes reduction in GCL thickness?

The finding that axonal damage precedes ganglion cell death was actually observed in patients affected by glaucoma (Reference 21). At present there is no demonstration in the literature that a similar phenomenon characterizes also patients affected by diabetic retinopathy (no published evidence that thinning of RNFL may precede the thinning of GCL in case of diabetes). We have therefore modified the phrase (page 9, lines 248-250) to explain that, at the moment, the evidence that axonal damage precedes ganglion cell death is restricted to glaucoma.

Reviewer 2 Report

The current study found that thinning of the retinal nerve fiber layer/ganglion cell layer (RNFL/GCL, the layer containing the retinal ganglion cells) precedes the death of these cells, implying that it could be a biomarker of diabetic retinopathy. I'd like to make the following observations.

1. Can the thinning of RNFL/GCL measured by OCT be identified in clinics in another non-invasive way? The biomarker will then be widely focused, as described in line 250.

2. In the current study, NGF (20.7±0.6 μm) treatment was effective. Please include a link to any previous report(s) that support this finding.

3. Another approach to dealing with this disorder appears to be to prevent diabetes-induced RNFL/GCL thinning.

Author Response

Reviewer 2 Point by point response

  1. Can the thinning of RNFL/GCL measured by OCT be identified in clinics in another non-invasive way? The biomarker will then be widely focused, as described in line 250.

At the moment, unfortunately, OCT represents the only methodology of use in clinics to measure the thickness of RNFL/GCL. Direct count of retinal ganglion cells can be performed only ex-vivo in murine and human retinas.

  1. In the current study, NGF (20.7±0.6 μm) treatment was effective. Please include a link to any previous report(s) that support this finding.

In the present study we demonstrated the efficacy of the treatment with eye drops NGF. A previous study performed by Hammes et al obtained similar results by systemic administration of NGF, this study is quoted in the manuscript as Reference 19

  1. Another approach to dealing with this disorder appears to be to prevent diabetes-induced RNFL/GCL thinning.

This is correct, in a previous study quoted in the manuscript as Reference 15 we demonstrated that topical treatment with NGF started at three weeks of age (the onset of diabetes in akita mice) allows to prevent the development of both diabetes driven neurodegeneration and vascular stage of DR in this animal model of diabetes. This preventive approach is certainly effective, but not easy to be translated into clinical practice as it would imply to treat with NGF all the diabetic patients (including children) starting from the onset of the disease, something very expensive and very complicated to be done, in particular when we know that only a subset of these patients are really predisposed to develop DR.

Reviewer 3 Report

Very interesting research - my objection refers to the relatively small number of mice in each group and to the personal, professional and scientific reserve of the topical effect of the drops in the neuroprotective effect on the retina. The methods and material of the work, the statistical analysis and the results of the work are correctly presented. Objectively written discussion.

Author Response

Reviewer 3 Point by point response

  1. Very interesting research - my objection refers to the relatively small number of mice in each group and to the personal, professional and scientific reserve of the topical effect of the drops in the neuroprotective effect on the retina. The methods and material of the work, the statistical analysis and the results of the work are correctly presented. Objectively written discussion.

Until few years ago, when performing ophthalmologic studies in animal models, a huge number of animals had to be used and sacrificed to perform analyses and experiments on the isolated retinas. The revolution came after the development of OCT instruments specific for rodents that allow to examine in vivo the retina of the animals and, when the instrument is positioned inside the animal facility, make it possible to study the same animals along time for the entire duration of the study. In this way it is possible to significantly reduce the number of animals to be studied (animals are sacrificed only at the end of the study and not at each time point).

The point concerning the effectiveness of eye drops NGF on the retina is well taken. Our study clearly shows that in mice the topical treatment works and the retina is protected. Will it be the same with humans? The study of Lambiase et al (Reference n.23 of this manuscript) support this hypothesis. Obviously, only a well designed and well performed clinical trial will allow to clarify this issue.